# Potential Correlation between Changes in Serum FGF21 Levels and Lenvatinib-Induced Appetite Loss in Patients with Unresectable Hepatocellular Carcinoma

**DOI:** 10.3390/cancers15123257

**Published:** 2023-06-20

**Authors:** Risako Kohya, Goki Suda, Masatsugu Ohara, Takashi Sasaki, Tomoka Yoda, Naofumi Sakurai, Sonoe Yoshida, Qingjie Fu, Zijian Yang, Shunichi Hosoda, Osamu Maehara, Shunsuke Ohnishi, Yoshimasa Tokuchi, Takashi Kitagataya, Kazuharu Suzuki, Naoki Kawagishi, Masato Nakai, Takuya Sho, Mitsuteru Natsuizaka, Koji Ogawa, Naoya Sakamoto

**Affiliations:** 1Department of Gastroenterology and Hepatology, Graduate School of Medicine, Hokkaido University, Sapporo 060-8648, Japan; lisakokgw@gmail.com (R.K.); masamasama_zu@yahoo.co.jp (M.O.); yume2made3ta@yahoo.co.jp (T.S.); tomokayoda.work@gmail.com (T.Y.); sakurai@pop.med.hokudai.ac.jp (N.S.); sonoeds@pop.med.hokudai.ac.jp (S.Y.); fuqingjie@pop.med.hokudai.ac.jp (Q.F.); yuang@eis.hokudai.ac.jp (Z.Y.); hosoda.shunichi.k0@elms.hokudai.ac.jp (S.H.); h.y.tokuchi112@med.hokudai.ac.jp (Y.T.); t.kitagataya@pop.med.hokudai.ac.jp (T.K.); kz.suzuki0524@gmail.com (K.S.); naopaleg@yahoo.co.jp (N.K.); mnakai@pop.med.hokudai.ac.jp (M.N.); shotaku@pop.med.hokudai.ac.jp (T.S.); mitsuteru1975@huhp.hokudai.ac.jp (M.N.); k-ogawa@med.hokudai.ac.jp (K.O.); sakamoto@med.hokudai.ac.jp (N.S.); 2Laboratory of Molecular and Cellular Medicine, Faculty of Pharmaceutical Sciences, Hokkaido University, Sapporo 060-8648, Japan; maeosa17@pharm.hokudai.ac.jp (O.M.); sonishi@pop.med.hokudai.ac.jp (S.O.)

**Keywords:** appetite loss, FGF21, hepatocellular carcinoma, lenvatinib, prognosis

## Abstract

**Simple Summary:**

Patients with unresectable hepatocellular carcinoma (HCC) who receive lenvatinib treatment often lose their appetite, which leads to poor outcomes, but the reason for appetite loss is unclear. FGF21 is a natural hormone-like factor that modulates appetite. We investigated the effect of lenvatinib on FGF21 levels in patients with HCC who were treated with lenvatinib and found that those with severe appetite loss showed increases in FGF21 levels before appetite loss occurred. This suggests that changes in FGF21 levels can be used to predict patients with a greater risk of marked appetite loss and provides insights into the mechanisms underlying lenvatinib-induced appetite loss in patients with HCC.

**Abstract:**

Lenvatinib, used for unresectable hepatocellular carcinoma (HCC), causes appetite loss, but the underlying mechanisms, clinical impact, and predictive factors have been unclear. The endocrine factor FGF21 modulates appetite and is involved in cachexia. We evaluated the association between FGF21 level changes during lenvatinib treatment for unresectable HCC and appetite loss. Sixty-three eligible unresectable HCC patients who started lenvatinib treatment between 2018 and 2021 were included. We analyzed FGF21 levels at baseline; 1, 2, and 4 weeks after lenvatinib initiation, and before the onset of appetite loss. Grade ≥ 2 lenvatinib-induced appetite loss led to liver functional reserve deterioration at disease progression and a poor prognosis. Baseline characteristics and serum FGF21 levels were similar between patients with and without appetite loss. However, the serum FGF21 change rate increased significantly at 4 weeks post-lenvatinib initiation in patients with grade ≥ 2 appetite loss, as compared to those without appetite loss. Similar significant increases in the serum FGF21 level change rate were observed prior to grade ≥ 2 appetite loss onset. This suggests that changes in FGF21 levels can be used to predict patients with a greater risk of marked appetite loss and provides insights into the mechanisms underlying lenvatinib-induced appetite loss in patients with HCC.

## 1. Introduction

Hepatocellular carcinoma (HCC) is a malignancy with a high mortality rate [1] and an increasing incidence [2]. Recently, because of successful clinical trials of novel anticancer drugs for unresectable HCC, various systemic therapies, including multikinase inhibitors of sorafenib [3], lenvatinib [4], regorafenib [5], and cabozantinib [6], a monoclonal antibody targeting vascular endothelial growth factor receptor (VEGFR) 2, ramucirumab [7], and a combination therapy involving the anti-VEGF-A antibody bevacizumab and the programmed death ligand 1 (PD-L1) inhibitor atezolizumab [8] have been approved. Real-world data have confirmed the efficacy and safety of those therapies [9,10,11,12,13,14]. Of these, atezolizumab and bevacizumab have been recommended as the first-line therapies in various guidelines because they have shown superior overall survival (OS) as compared with the previous standard therapy of sorafenib [8]. Lenvatinib has been recommended as a first-line therapy in patients with advanced HCC who are not indicated for atezolizumab and bevacizumab and has also been recommended as a second-line systemic therapy for patients with advanced HCC [15].

Lenvatinib is a multikinase inhibitor that targets various signaling pathways, including VEGFR-1–3, c-Kit, and fibroblast growth factor receptors (FGFR)-1–4 [16,17]. In a phase 3 clinical trial involving lenvatinib for unresectable HCC (REFLECT trial), lenvatinib was shown to be non-inferior to sorafenib in terms of OS, and the median PFS with lenvatinib was significantly longer than that with sorafenib [4]. In this clinical trial, the lenvatinib-treated group showed a higher rate of grade ≥ 3 adverse events (AEs) in terms of decreased appetite (5% vs. 1% with lenvatinib and sorafenib), body-weight loss (8% vs. 3% with lenvatinib and sorafenib), and any grade AEs regarding nausea (20% vs. 14% with lenvatinib and sorafenib) and vomiting (16% vs. 8% with lenvatinib and sorafenib).

Grade ≥ 2 or higher appetite loss and related AEs are clinically important because real-world data have revealed that, in patients with unresectable HCC, appetite loss after lenvatinib treatment was associated with discontinuation of lenvatinib [18]. In addition, grade ≥ 2 appetite loss requires dose reduction or discontinuation of lenvatinib treatment according to the lenvatinib package insert. However, the mechanisms underlying appetite loss and related symptoms induced by lenvatinib have not been elucidated. Consequently, an optimal approach for managing these AEs has not yet been clearly defined.

We and other groups have reported that levels of fibroblast growth factors (FGFs), including FGF19 and FGF23, significantly increased after lenvatinib treatment for unresectable HCC [12,19]. Fibroblast growth factor 21 (FGF21) is a member of the FGF19 subfamily, which also includes FGF19 and FGF23. FGF21 acts as an endocrine factor [20], binds FGFR 1c, FGFR2c, or FGFR3c with the co-receptor klotho, and activates FGFR-mediated signaling [21]. In addition, FGF21 is a key regulator of metabolism and an appetite modulator [21,22]. FGF21 exerts effects on metabolism and appetite via direct FGF21 signaling to the central nervous system, which expresses FGFRs and β-klotho [21,22]. However, very low 14C-lenvatinib distribution to the brain has been reported in the administration of 14C-lenvatinib to rats (https://www.accessdata.fda.gov/drugsatfda_docs/nda/2015/206947Orig1s000PharmR.pdf, accessed on 21 May 2023). Thus, the ability of lenvatinib to pass through the blood–brain barrier (BBB) is thought to be low. Therefore, lenvatinib may not have a strong suppressive effect on FGF-mediated signaling in the central nervous system. Thus, we reasoned that, if FGF21 levels increase after lenvatinib administration, elevated FGF21 levels might affect appetite and metabolism.

Consequently, we here analyzed the prevalence of appetite loss, particularly grade ≥ 2 appetite loss, the effect of appetite loss on PFS, OS, liver functional reserve changes, and serum albumin, and the association between lenvantinib-related changes in FGF21 levels and the AE of appetite loss.

## 2. Materials and Methods

### 2.1. Patients and Study Design

In this retrospective study, eligible patients with unresectable HCC who were treated with lenvatinib at Hokkaido University Hospital between April 2018 and October 2021 were screened. Patients were included if they were followed-up for more than 2 months, had properly preserved serum for analyzing FGF21, were evaluated for treatment response using dynamic computed tomography (CT) every 2–3 months, and had sufficient clinical data, including AEs of lenvatinib. We excluded patients if they did not have sufficient clinical data, including lenvatinib-induced AEs, sufficiently preserved serum for evaluating FGF21, or were treated with other anti-cancer therapies simultaneously.

We collected baseline laboratory data, including tumor markers, sex, age, HCC etiology, Barcelona Clinic Liver Cancer (BCLC) stage, and Child–Pugh liver functional reserve. Adverse events were determined following the guidelines of the Common Terminology Criteria for Adverse Events (CTCAE), version 5.0. We analyzed FGF21 levels at baseline, at 1–2 weeks, and at 4 weeks after lenvatinib initiation. In addition, if available, we evaluated serum FGF21 levels around the onset of grade ≥ 2 appetite loss (within 2 weeks) and ahead of the onset of grade ≥ 2 appetite loss (within 2 weeks). We further investigated the association between changes in serum FGF21 levels and AEs associated with appetite loss. All included patients were evaluated for treatment response using the modified Response Evaluation Criteria in Solid Tumors (mRECIST) based on dynamic CT [23].

The study protocol was approved by the Hokkaido University Hospital Ethics Committee (approval number: 017–0521). Patients who provided written informed consent or did not decline to participate were included in this study. The ethics committee approved the inclusion of patients who did not decline to participate in the study, instead of obtaining written informed consent from the included patients. The study conformed to the ethical guidelines of the Declaration of Helsinki.

### 2.2. Analysis of FGF21

Serum FGF21 levels were analyzed using commercial enzyme-linked immunosorbent assays (R&D Systems, Minneapolis, MN, USA) according to the manufacturer’s protocols [24].

### 2.3. Lenvatinib Treatment Protocol

Lenvatinib was administered orally once daily at a dose of 8 or 12 mg to patients weighing < 60 or ≥60 kg, respectively. Lenvatinib was discontinued if disease progression or AEs were observed. The lenvatinib dose was adjusted by the attending physician based on AEs and tolerability [19].

### 2.4. Statistical Analysis

Categorical variables were analyzed using chi-squared and Fisher’s exact tests. Continuous variables were analyzed using Student’s *t*-test or the Mann–Whitney U test. We compared the differences among three or more populations using a one-way analysis of variance followed by Tukey’s test. Survival curves were analyzed using the Kaplan–Meier method and were compared using the log-rank test at the 12-month time-point. Optimal cut-off values of the rate of changes in FGF21 for predicting grade ≥ 2 appetite loss during lenvatinib treatment were set based on the receiver operating characteristic (ROC) curve results, by maximizing the Youden index. A multivariate logistic regression analysis with stepwise forward selection was performed, using variables identified at *p* ≤ 0.15 in univariate analyses of the baseline factors associated with the occurrence of grade ≥ 2 appetite loss during lenvatinib and the changes in FGF21 between baseline and 4 weeks post-lenvatinib initiation. Statistical analyses were performed using Prism 9.41 (GraphPad Software, La Jolla, CA, USA).

## 3. Results

### 3.1. Patient Enrollment and Baseline Characteristics

We screened 70 consecutive patients with unresectable HCC who were treated with lenvatinib at the Hokkaido University Hospital between April 2018 and October 2021. Of these patients, seven were excluded from the analysis because they had insufficient clinical data and/or lacked sufficient preserved serum for analysis of serum FGF21 levels. Ultimately, 63 patients with unresectable HCC were included in this study (Figure 1). Table 1 shows the baseline characteristics of the patients. The median age of the included patients was 70 years (range, 45–88 years), and 90.5% (57/63) were males. Forty-six and 17 patients had Child–Pugh grades A and B, respectively, and one, 29, and 33 patients had BCLC stages A, B, and C, respectively. Median serum albumin levels were 3.7 g/dL (range, 2.6–4.6 g/dL).

Overall, 12 (19.0%), 23 (36.5%), 22 (34.9%), and 6 (9.5%) patients experienced the best response of complete response, partial response, stable disease, and progressive disease, respectively.

### 3.2. Occurrence Rate of Appetite Loss during Lenvatinib Treatment and Comparison between Patients with or without Grade ≥ 2 Appetite Loss

The occurrence rates of grade 1 and grade 2 or higher appetite loss during lenvatinib treatment were 46% (29/63) and 23% (14/63), respectively (Table 2).

As shown in Table 3, the baseline patient characteristics were similar between patients who did and did not experience grade ≥ 2 appetite loss during lenvatinib treatment. The median onset of appetite loss was 21.5 days (range, 2–532 days) after lenvatinib initiation in all patients with appetite loss. In addition, the median onset of grade ≥ 2 appetite loss occurred at 28 days (range 7–532 days) after lenvatinib initiation.

### 3.3. Impact of Grade ≥ 2 Appetite Loss on Treatment Response, Prognosis, and Liver Functional Reserve

Subsequently, we analyzed the effect of appetite loss on PFS, OS, changes in liver functional reserve, and serum albumin levels. As shown in Figure 2A, median OS was significantly shorter in patients who experienced grade ≥ 2 appetite loss than in those who did not experience such loss (median OS, 15.00 months vs. 8.50 months; hazard ratio [HR]: 2.763, 95% confidence interval [CI]: 1.03–7.45, *p* = 0.007). On the other hand, as shown in Figure 2B, the PFS was similar between patients who did and did not experience grade ≥ 2 appetite loss during lenvatinib treatment.

To clarify the reason for the significant reduction in median OS observed in patients experiencing grade ≥ 2 appetite loss as compared to those who did not, despite similar PFS between the two groups, we analyzed the changes in liver functional reserve at the point of disease progression. As depicted in Figure 3A, patients who experienced grade ≥ 2 appetite loss demonstrated a significant deterioration in Child–Pugh scores as compared to those who did not experience appetite loss. This suggests that a grade ≥ 2 appetite loss may lead to a decline in the liver functional reserve. Additionally, as illustrated in Figure 3B, the proportion of patients who were unable to undergo salvage chemotherapy following lenvatinib treatment failure was marginally higher among those who experienced appetite loss than among those who did not. 

### 3.4. Changes in Serum FGF21 Levels during Lenvatinib Treatment and the Relationship between Changes in Serum FGF21 Levels and Appetite Loss

Next, we analyzed changes in serum FGF21 levels during lenvatinib treatment in patients with unresectable HCC. As depicted in Figure 4A, the median baseline serum FGF21 level was 411 pg/mL (range, 149–3224 pg/mL). Figure 4B shows that the rate of change in serum FGF21 levels did not change significantly until 4 weeks after lenvatinib initiation. We then compared the rate of change in FGF21 levels between patients who experienced grade ≥ 2 appetite loss and those who did not, considering that the median onset of grade ≥ 2 appetite loss occurred at 28 days (range 7–532 days) after starting lenvatinib. As shown in Figure 4C, the rate of change in FGF21 levels was significantly higher in patients who experienced grade ≥ 2 appetite loss than in those who did not experience appetite loss at 4 weeks after lenvatinib initiation (*n* = 63).

Subsequently, we compared the rate of change in serum FGF21 levels in samples obtained within a period of 14 days before and after the onset of appetite loss. For patients who did not experience appetite loss, the rate of change in FGF21 levels was analyzed between baseline and 4 weeks after lenvatinib initiation, which was close to the median onset of appetite loss in this study. As depicted in Figure 4D, the rate of change in FGF21 levels around the time of onset of appetite loss was significantly higher in patients who then experienced grade ≥ 2 appetite loss than in those who did not (*n* = 58).

Finally, we analyzed the rate of change in FGF21 levels in serum samples obtained within 14 days prior to the onset of appetite loss (*n* = 47). As shown in Figure 4E, the rate of change in FGF21 prior to the onset of appetite loss was significantly increased in patients who experienced grade ≥ 2 appetite loss as compared to those who did not experience appetite loss.

Next, we analyzed optimal cut-off values of the rate of change in FGF21 levels for predicting grade ≥ 2 appetite loss during lenvatinib treatment. As shown in Appendix A, the cut-off value of the rate of change in FGF21 levels at 4 weeks after lenvatinib treatment initiation predicting grade ≥ 2 appetite loss was set at 0.174 (sensitivity 85.7, specificity 57.1; area under the ROC curve, 0.66 [95%CI 0.46–0.858]). Subsequently, we conducted a multivariate logistic regression analysis regarding the factors associated with the occurrence of grade ≥ 2 appetite loss during lenvatinib. Multivariate analysis revealed that the rate of changes in FGF21 at 4 weeks after lenvatinib were significantly associated with the occurrence of grade ≥ 2 appetite loss during lenvatinib (odds ratio 1.010 (95%CI 1.001–1.020), *p* = 0.037) (Appendix A).

Furthermore, because to the best of our knowledge, the direct relationship between the changes in FGF21 and serum albumin levels have been fully elucidated to date, we analyzed the relationship between the changes in serum albumin and FGF21 levels 4 weeks after lenvatinib initiation. As depicted in Figure 5, patients who experienced an increased rate of change in FGF21 levels at 4 weeks after lenvatinib initiation had a significantly decreased median serum albumin level as compared with those who did not experience an increase in FGF21 at the same time point.

## 4. Discussion

In this study, we found that grade ≥ 2 appetite loss induced by lenvatinib for unresectable HCC led to the deterioration of the liver functional reserve at the time of disease progression. Additionally, patients who experienced appetite loss were less likely to receive salvage therapy after lenvatinib failure or intolerance. Furthermore, we discovered that grade ≥ 2 appetite loss induced by lenvatinib resulted in a poor prognosis. This highlighted that identifying the factors associated with appetite loss is a crucial clinical issue. In this study, although the baseline characteristics were similar between patients with and without appetite loss, we discovered that the rate of change in serum FGF21 levels increased significantly at 4 weeks after lenvatinib initiation in patients as compared to those without grade ≥ 2 appetite loss. These significant increases in the rate of change of serum FGF21 levels were similarly observed prior to the onset of grade ≥ 2 appetite loss. These results suggested that changes in serum FGF21 levels during lenvatinib treatment are associated with appetite loss, providing mechanistic insight into the development of appetite loss during lenvatinib treatment. Furthermore, changes in serum FGF21 levels may serve as a predictive factor for severe appetite loss during lenvatinib treatment.

In this study, we found that the median onset of appetite loss occurred 21.5 days after the initiation of lenvatinib treatment, which is consistent with the results of a previous study [18]. Although the overall rate of change in serum FGF21 levels was similar at baseline and at 1, 2, and 4 weeks after lenvatinib initiation, we observed a significant increase in the rate of change in serum FGF21 levels at 4 weeks after lenvatinib initiation in patients with grade ≥ 2 appetite loss as compared to those without it. Moreover, we similarly observed a significant increase in the rate of change in serum FGF21 levels in patients with than in patients without grade ≥ 2 appetite loss before the onset of such appetite loss. The relationship between serum FGF21 levels before and after appetite loss onset and appetite loss is not well understood; however, several hypotheses have been proposed. First, it is thought that FGF21 is an endocrine FGF [20,21] that can modulate metabolism and appetite via the central nervous system [22,25]. Thus, elevated FGF21 levels could directly affect the central nervous system, resulting in a moderation of appetite after lenvatinib treatment. Although lenvatinib could suppress FGFR1–4-mediated signaling [16], most lenvatinib could not pass through the BBB, resulting in low suppression of FGFR signaling by lenvatinib in the central nervous system.

The effect of elevated FGF21 levels on appetite in the context of advanced malignancy and poor functional liver reserve has not been well elucidated. Franz et al. reported that serum FGF21 levels were significantly increased in patients with cachexia [26]. This supports the hypothesis that lenvatinib-mediated increases in FGF21 levels contribute to appetite loss and decreased body weight [26]. However, further analyses are required.

A second hypothesis regarding the cause of elevated FGF21 levels before the onset of appetite loss in patients with grade ≥ 2 appetite loss after lenvatinib treatment for unresectable HCC is that a fasting or pre-cachexia status may be established before the onset of severe appetite loss in treated patients [27,28], resulting in the elevation of FGF21 levels before the onset of severe appetite loss. However, it typically takes a longer time to increase FGF21 levels during fasting, as FGF21 levels do not change during short fasting periods in humans [29].Further analysis is required to confirm this hypothesis.

In this study, we revealed that, unlike other FGF19 family members, including FGF19 and FGF23 [12,19,20], the median serum FGF21 levels did not change during lenvatinib treatment. The reason for the lack of change in the median levels of only FGF21 among the FGF19 family members has not yet been clarified. It may be because FGF21 is mainly produced by hepatocytes, while FGFR1 is rarely expressed in normal human hepatocytes [30]; thus, FGF21 does not act on normal human hepatocytes. Consequently, compensatory FGF21 elevation might not occur after suppressing FGF21-mediated signaling by lenvatinib in normal human hepatocytes. HepG2 human hepatoma cell lines express FGFR1 at high levels. Moreover, FGF21 can affect HepG2 cells directly and alter the production of triglycerides. Thus, in particular HCC subtypes, lenvatinib may cause an elevation of FGF21 [31,32]. However, further analyses are required to validate this hypothesis.

In this study, we conducted ROC curve analysis and a multivariate analysis of factors linked with grade 2 or greater appetite loss during the course of lenvatinib treatment for unresectable HCC. The AUC was calculated to be 0.66, with a sensitivity of 85.7% and a specificity of 57.1%. The multivariate analysis revealed that changes in FGF21 rates 4 weeks after initiating lenvatinib treatment were correlated with the occurrence of grade 2 or more severe appetite loss during lenvatinib treatment (Odds Ratio, 1.010). Considering the relatively low levels of AUC and odds ratio, it would be necessary to include additional predictive factors for appetite loss during lenvatinib treatment to improve the accuracy of the prediction model. Therefore, further analysis is warranted.

Lenvatinib has been widely used in the treatment of various malignancies [33,34] and appetite loss is a common side effect of lenvatinib in various malignancies. Therefore, it is worthwhile investigating the relationship between appetite loss and elevated FGF21 levels following lenvatinib treatment.

In this study, we found that grade ≥ 2 appetite loss was associated with a poor prognosis. This could be due to appetite loss, causing deterioration of the liver functional reserve, which in turn hindered salvage therapy after lenvatinib treatment. Additionally, elevated FGF21 [35] and decreased food intake could cause a loss of muscle mass, as sarcopenia in patients with chronic liver disease [36,37,38] is an independent factor that contributes to poor prognosis. It might also affect the poor patient prognosis observed in this study.

This study had several limitations, including its retrospective design and a relatively small number of included patients. In addition, several data, including the data on alcohol intake during lenvatinib treatment, could not be collected due to the retrospective nature of the study. Therefore, further large-scale prospective studies are required to validate these findings.

## 5. Conclusions

In this study, we revealed that grade ≥ 2 appetite loss induced by lenvatinib treatment for unresectable HCC resulted in a poor prognosis. Additionally, we found that the rate of change in serum FGF21 levels increased significantly before the onset of grade ≥ 2 appetite loss. These results provided mechanistic insights into the development of appetite loss during lenvatinib treatment, and suggest that changes in serum FGF21 levels might serve as a predictive factor for severe appetite loss during lenvatinib treatment.

## Figures and Tables

**Figure 1 cancers-15-03257-f001:**
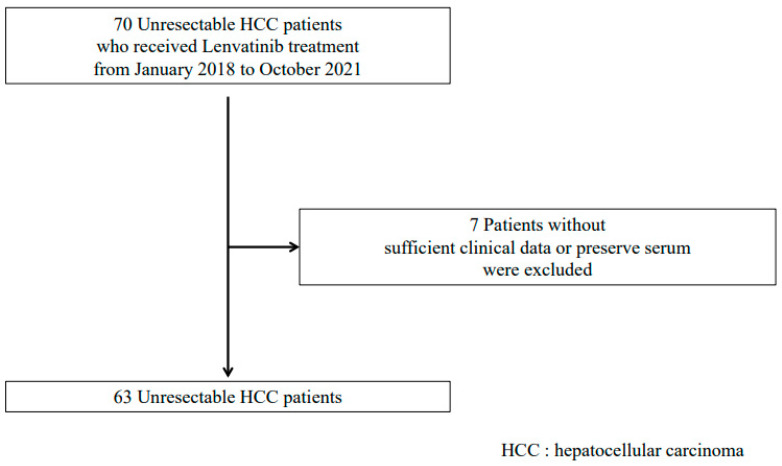
Study flow.

**Figure 2 cancers-15-03257-f002:**
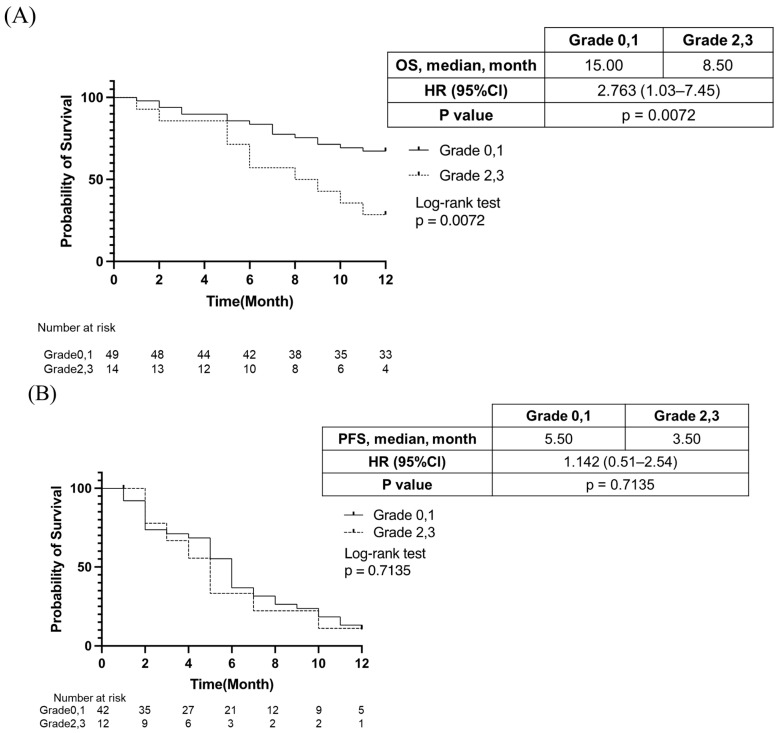
(**A**). Comparison of overall survival (OS) between patients with or without grade ≥ 2-grade appetite loss. (**B**). Comparison of progression-free survival (PFS) between patients with or without grade ≥ 2-grade appetite loss.

**Figure 3 cancers-15-03257-f003:**
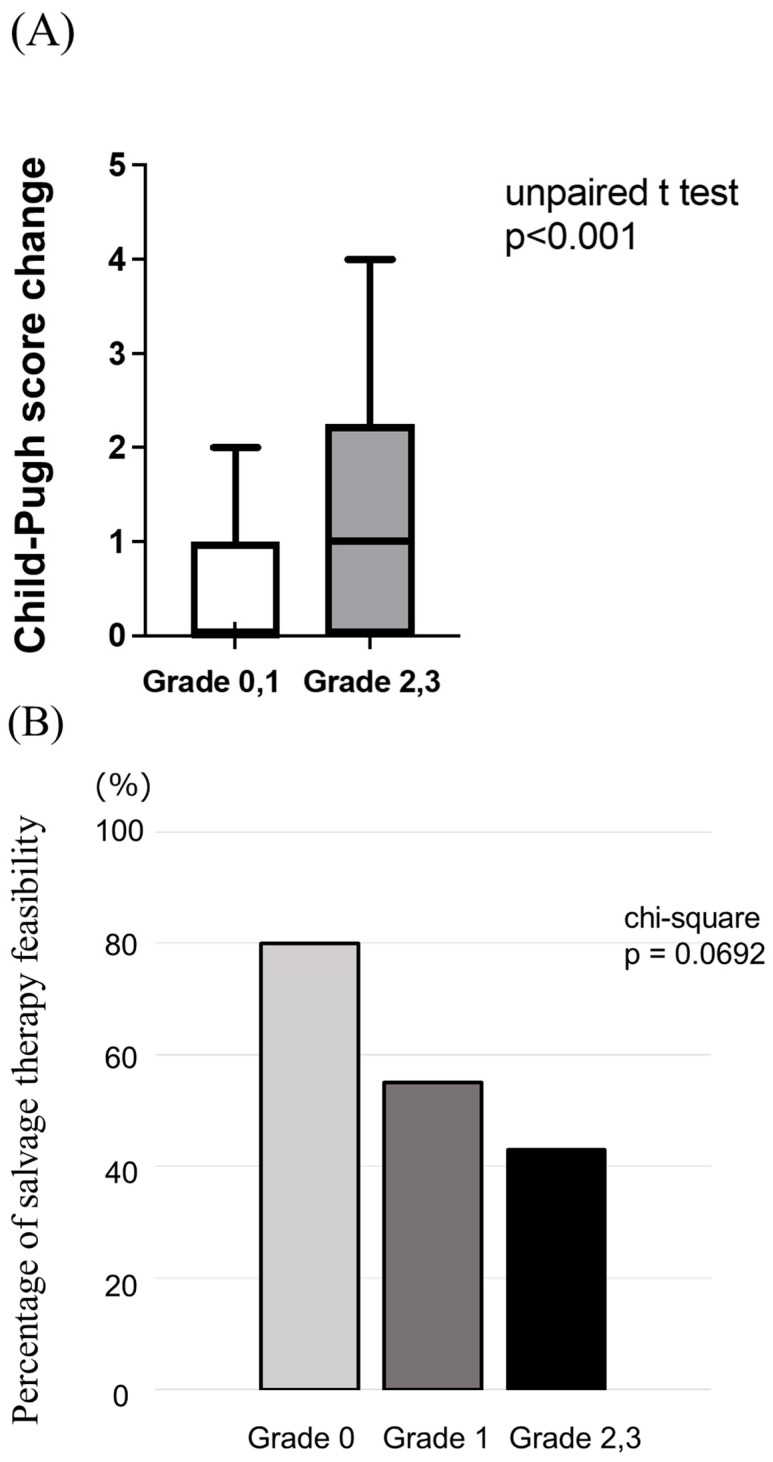
(**A**). Comparison of changes in liver functional reserve between patients with or without appetite loss. (**B**). Comparison of the rate of cases in whom salvage therapy was possible between patients with or without appetite loss.

**Figure 4 cancers-15-03257-f004:**
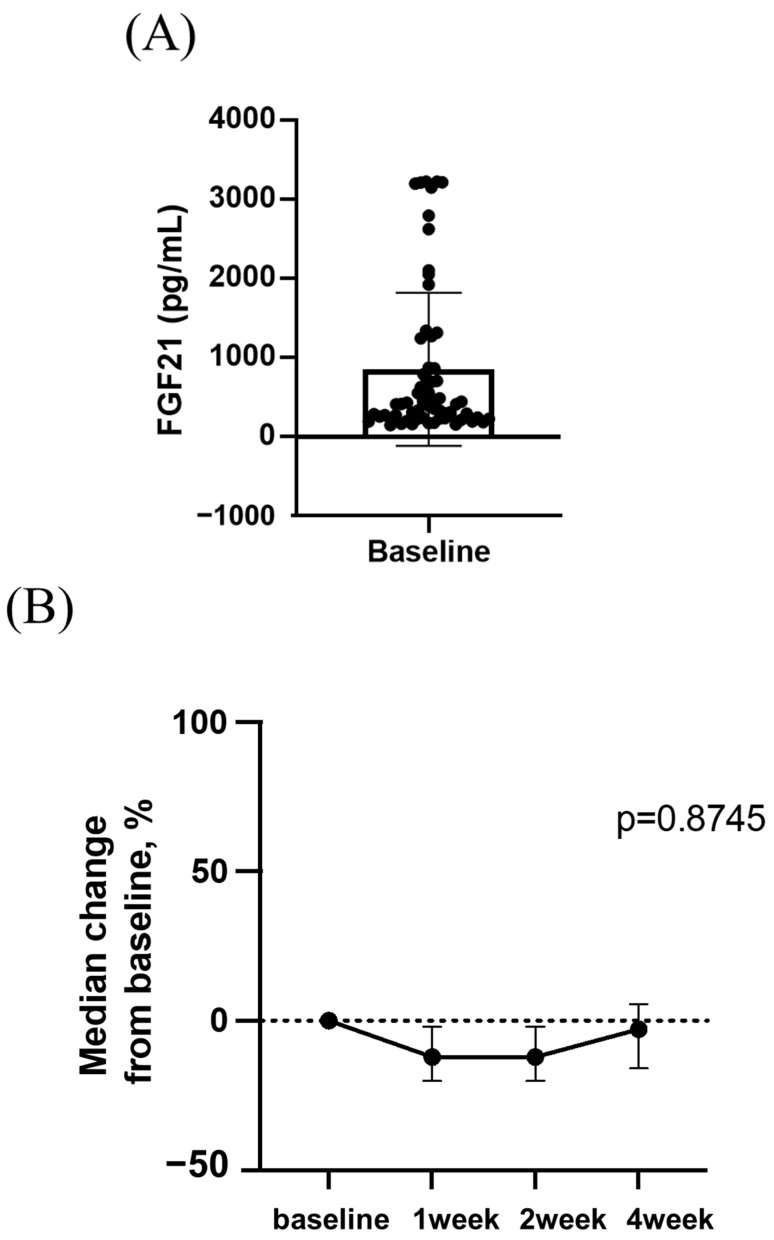
(**A**). Baseline serum FGF21 levels. (**B**). Changes in serum FGF21 levels in the whole cohort. (**C**). Comparison of the rate of change in serum FGF21 levels between patients with or without grade ≥ 2-grade appetite loss at 4 weeks after lenvatinib initiation. (**D**). Comparison of the rate of change in serum FGF21 levels between patients with or without grade ≥ 2-grade appetite loss within a period of 14 days before and after the onset of appetite loss. (**E**). Comparison of the rate of change in serum FGF21 levels between patients with or without grade ≥ 2 appetite loss within a period of 14 days ahead of appetite loss onset (*n* = 47).

**Figure 5 cancers-15-03257-f005:**
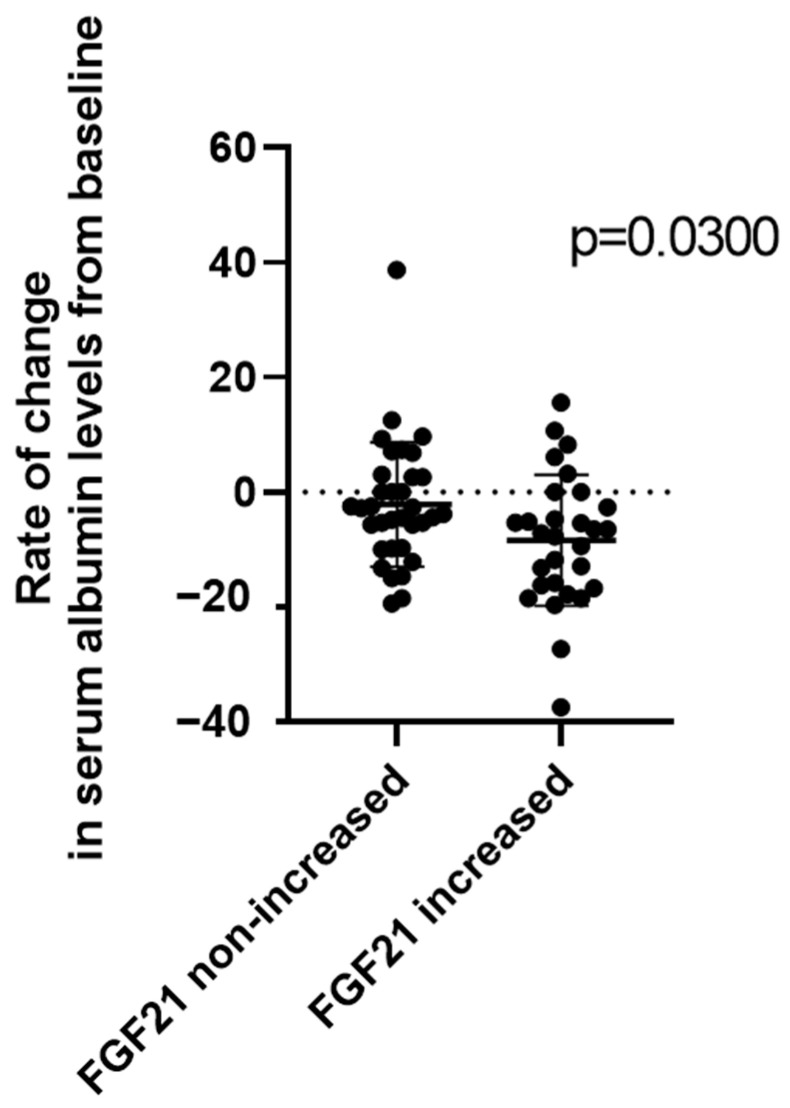
Comparison of changes in serum albumin levels between patients with or without increased serum FGF21 levels at 4 weeks after lenvatinib initiation.

**Table 1 cancers-15-03257-t001:** Baseline patient characteristics.

Age (years), median (range)	70 (45–88)
Sex (Male/Female)	57/6
BMI (kg/m^2^)	24 (17–49)
ECOG performance status (0/1)	47/16
History of diabetes mellitus (0/1)	29/34
Etiology, *n* (%)	
HBV	20 (31.7%)
HCV	11 (17.4%)
NBNC	32 (50.7%)
Child–Pugh class, *n* (%)	
A	46 (73.0%)
B	17 (27.0%)
BCLC stage, *n* (%)	
A	1 (1.6%)
B	29 (46.0%)
C	33 (52.4%)
TNM classification, *n* (%)	
2	8 (12.7%)
3	23 (36.5%)
4A	19 (30.2%)
4B	13 (20.6%)
Biochemical analysis, median (range)	
Platelet, ×10^4^/μL	16.1 (4.4–51.7)
Prothrombin time, %	88.3 (40.7–117.7)
Serum albumin, g/dL	3.7 (2.6–4.6)
AST, IU/L	38.0 (15.0–303.0)
ALT, IU/L	24.0 (8.0–168.0)
Total bilirubin, mg/dL	0.8 (0.3–3.1)
Alpha-fetoprotein, ng/mL	78.2 (1.6–449,909.0)
Alpha-fetoprotein L3, ng/mL	27.2 (0.5–99.5)

HBV, hepatitis B virus; HCV, hepatitis C virus; NBNC, non-B non-C; BCLC, Barcelona Clinic Liver Cancer; AST, aspartate aminotransferase; ALT, alanine aminotransferase; AFP, alpha-fetoprotein; PIVKA-II, induced by vitamin K absence or antagonist-II; FGF, fibroblast growth factor, ANG2: angiopoietin 2; VEGF, vascular endothelial growth factor.

**Table 2 cancers-15-03257-t002:** Prevalence of patients with appetite loss during lenvatinib treatment for unresectable HCC (*n* = 63).

Appetite loss	
Any grade	43 (68%)
Grade 1	29 (46%)
Grade 2	13 (21%)
Grade 3	1 (2%)

HCC: Hepatocellular carcinoma.

**Table 3 cancers-15-03257-t003:** Comparison of patient characteristics between patients with and without grade ≥ 2 appetite loss.

	Grade 0, 1 (N = 49, 78%)	Grade 2, 3 (N = 14, 22%)	*p* Value
Age (years), median (range)	70 (45–82)	72 (54–88)	
Sex (Male/Female)	45/4	12/2	0.252
BMI (kg/m_2_)	24 (17–34)	25 (18–49)	0.607
ECOG Performance status (0/1)	37/12	10/4	0.136
History of Diabetes mellitus (0/1)	21/28	6/8	0.739
Etiology, n (%)			>0.999
HBV	16	4	
HCV	10	1	0.732
NBNC	23	10	
Child–Pugh class, n (%)			
A	37	9	
B	12	5	0.498
BCLC stage, n (%)			
A	0	1	
B	22	7	0.732
C	27	6	
TNM classification, n (%)			
2	5	2	
3	19	3	0.144
4A	14	5	
4B	11	3	
Biochemical analysis, median (range)			
Total bilirubin, mg/dL	0.8 (0.3–3.1)	0.8 (0.4–2.6)	
Platelets. ×104/μL	16.0 (4.4–51.7)	18.5 (6.5–29.5)	0.733
Prothrombin time, %	88.4 (40.7–117.7)	87.1 (56.5–116.8)	0.763
Serum albumin, g/dL	3.8 (2.7–4.6)	3.7 (2.6–4.6)	0.792
AST, IU/L	37.0 (15.0–303.0)	38.50 (18.0–153.0)	0.275
ALT, IU/L	26.0 (8.0–168.0)	23.5 (13.0–48.0)	0.573
Alpha-fetoprotein, ng/mL	92.6 (1.6–449909.0)	12.5 (2.2–257916.0)	0.165
ALBI grade			0.451
1	17	2	
2	29	10	0.262
3	3	2	

HBV, hepatitis B virus; HCV, hepatitis C virus; NBNC, non-B non-C; BCLC, Barcelona Clinic Liver Cancer; AST, aspartate aminotransferase; ALT, alanine aminotransferase; AFP, alpha-fetoprotein; PIVKA-II, induced by vitamin K absence or antagonist-II; FGF, fibroblast growth factor, ANG2: angiopoietin 2; VEGF, vascular endothelial growth factor; ALBI, albumin–bilirubin.

## Data Availability

All data generated or analyzed during this study are included in this article. Further inquiries can be directed to the corresponding authors.

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
