# Peer review of "Potential Correlation between Changes in Serum FGF21 Levels and Lenvatinib-Induced Appetite Loss in Patients with Unresectable Hepatocellular Carcinoma"

_cancers, 2023, doi:10.3390/cancers15123257_

Round 1

Reviewer 1 Report

Authors describe how serum FGF21 level is responsible for Lenvatinib induced appetite loss in unsecectable HCC patients. Overall, the manuscript is well written and well structured. Though I have few concerns.

Authors never mentioned the full name of FGF21. I think they supposed to mentioned at least one. 

It is quite established that FGF21 is related to appetite loss. But this study uniquely showed that serum FGF21 is related with Lenvatinib induced appetite suppression. In my opinion the study will be more interesting if they can identify why FGF21 level is changing in some patients with Lenvatinib treatment but not for others. Are there any gene mutations in those patients?

It is obvious that serum albumin level will decrease when body is unable to absorb nutrients.  It's well known that Lenvatinib cause appetite loss. So, I didn't understand what they want to prove to show comparison of changes in serum albumin levels between patients with or without increased serum FGF21 levels at 4 weeks after Lenvatinib initiation.

Discussion can be more concise and simple.

I have no issues with the quality of English language.

Author Response

June 11, 2023

Prof. Dr. Samuel C. Mok

Editor-in-Chief

Cancers

Dear Editor:

We are grateful for your consideration of our manuscript titled “Potential correlation between changes in serum FGF21 levels and lenvatinib-induced appetite loss in patients with unre-sectable hepatocellular carcinoma.

We greatly appreciate the comments of the editor and reviewers and addressing them has significantly improved the quality of the manuscript. We hope that the revised manuscript is now suitable for publication in Cancers.

Our point-by-point responses to the editor and reviewers’ comments are attached below. Changes to the text are indicated in red for ease of review.

Thank you for your consideration. I look forward to hearing from you.

Sincerely,

Goki Suda, M.D., Ph.D

Department of Gastroenterology and Hepatology/Graduate School of Medicine

Hokkaido University, North 15, West 7, Kita-ku,

Sapporo, Hokkaido 060-8638, Japan

Phone number: +81 11-716-1161

Fax number: +81 11-706-7867

Email address: [email protected]

Responses to Reviewers

Reviewer 1

Comments and Suggestions for Authors

Authors describe how serum FGF21 level is responsible for Lenvatinib induced appetite loss in unsecectable HCC patients. Overall, the manuscript is well written and well structured. Though I have few concerns.

 Response

We sincerely appreciate the reviewer’s helpful and constructive comments and suggestions. We have addressed each of these in detail below.

Authors never mentioned the full name of FGF21. I think they supposed to mentioned at least one.

Response 

We sincerely appreciate the reviewer’s suggestion. We have addressed this in line 73.

It is quite established that FGF21 is related to appetite loss. But this study uniquely showed that serum FGF21 is related with Lenvatinib induced appetite suppression. In my opinion the study will be more interesting if they can identify why FGF21 level is changing in some patients with Lenvatinib treatment but not for others. Are there any gene mutations in those patients?

Response

We sincerely appreciate the reviewer’s helpful and constructive comment. As pointed out by the reviewer, the reason for changes in FGF21 levels in some patients undergoing lenvatinib treatment but not in others is not clear, although some hypotheses have been put forward. In this study, we revealed that, unlike other FGF19 family members, including FGF19 and FGF23 [1-3], the median serum FGF21 levels did not change during lenvatinib treatment. The reason for the lack of change in the median levels of only FGF21 among the FGF19 family members has not yet been clarified. It may be because FGF21 is mainly produced by hepatocytes, while FGFR1 is rarely expressed in normal human hepatocytes [4]; thus, FGF21 does not act on normal human hepatocytes. Consequently, compensatory FGF21 elevation might not occur after suppressing FGF21-mediated signaling by lenvatinib in normal human hepatocytes, whereas human hepatoma cell lines of HepG2 express FGFR1 at high levels. Moreover, FGF21 can affect HepG2 cells directly and alter the production of triglycerides [5,6]. Thus, in particular HCC subtypes with genetic alteration, lenvatinib may cause elevation of FGF21. However, further analyses are required to validate this hypothesis. We have described this in lines 313–324.

It is obvious that serum albumin level will decrease when body is unable to absorb nutrients.  It's well known that Lenvatinib cause appetite loss. So, I didn't understand what they want to prove to show comparison of changes in serum albumin levels between patients with or without increased serum FGF21 levels at 4 weeks after Lenvatinib initiation.

Response

We sincerely appreciate the reviewer’s helpful and constructive comments and suggestions. We completely agree that appetite loss would result in decreased serum albumin levels. While, to the best of our knowledge, the direct relationship between the changes in FGF21 and serum albumin levels have been not clarified to date. We have added this description in lines 258–259.

Discussion can be more concise and simpler.

 Response

We sincerely appreciate the reviewer’s suggestion. As pointed out by the reviewer, we have revised and deleted some sentences in the Discussion to simplify it and make it more concise.

We greatly appreciate the comments of the reviewers and addressing them has significantly improved the quality of the manuscript. We hope that the revised manuscript is now suitable for publication in Cancers.

Reviewer 2 Report

Potential correlation between changes in serum FGF21 levels and Lenvatinib induced appetite loss in patients with unresec- 3 table hepatocellular carcinoma

The present study evaluates the association between FGF21 level changes during lenvatinib treatment for unresectable HCC and appetite loss. The study highlights the prognostic utility of FGF21 to predict lenvatinib-induced appetite loss in patients with HCC. The approach and the overall design of the study are good.  The manuscript is well written. Future implications of the study should be added.

Author Response

June 11, 2023

Prof. Dr. Samuel C. Mok

Editor-in-Chief

Cancers

Dear Editor:

We are grateful for your consideration of our manuscript titled “Potential correlation between changes in serum FGF21 levels and lenvatinib-induced appetite loss in patients with unre-sectable hepatocellular carcinoma.

We greatly appreciate the comments of the editor and reviewers and addressing them has significantly improved the quality of the manuscript. We hope that the revised manuscript is now suitable for publication in Cancers.

Our point-by-point responses to the editor and reviewers’ comments are attached below. Changes to the text are indicated in red for ease of review.

Thank you for your consideration. I look forward to hearing from you.

Sincerely,

Goki Suda, M.D., Ph.D

Department of Gastroenterology and Hepatology/Graduate School of Medicine

Hokkaido University, North 15, West 7, Kita-ku,

Sapporo, Hokkaido 060-8638, Japan

Phone number: +81 11-716-1161

Fax number: +81 11-706-7867

Email address: [email protected]

Responses to Reviewers

Reviewer 2

Comments and Suggestions for Authors

Potential correlation between changes in serum FGF21 levels and Lenvatinib induced appetite loss in patients with unresec- 3 table hepatocellular carcinoma

The present study evaluates the association between FGF21 level changes during lenvatinib treatment for unresectable HCC and appetite loss. The study highlights the prognostic utility of FGF21 to predict lenvatinib-induced appetite loss in patients with HCC. The approach and the overall design of the study are good.  The manuscript is well written. Future implications of the study should be added. 

Response

We sincerely appreciate the reviewer’s helpful and constructive comments and suggestions. As pointed out by the reviewer, we conducted further analysis to enforce the results.

We added these descriptions in lines 130–136, lines 248–257, and Supplementary Figure S1 and supplementary table S1.

“Next, we analyzed optimal cut-off values of the rate of change in FGF21 levels for predicting grade ≥ 2 appetite loss during lenvatinib treatment. As shown in Supplementary Figure S1, the cut-off value of the rate of change in FGF21 levels at 4 weeks after lenvatinib treatment initiation predicting grade ≥ 2 appetite loss was set at 0.174 (sensitivity 85.7, specificity 57.1; area under the ROC curve, 0.66 [95%CI 0.46–0.858]). Subsequently, we conducted a multivariate logistic regression analysis regarding the factors associated with occurrence of grade ≥ 2 appetite loss during lenvatinib. Multivariate analysis revealed that the rate of changes in FGF21 at 4 weeks after lenvatinib were significantly associated with occurrence of grade ≥ 2 appetite loss during lenvatinib (odds ratio 1.010 (95%CI 1.001–1.020), p = 0.037) (Supplementary Table S1).”

We greatly appreciate the comments of the editor and reviewers and addressing them has significantly improved the quality of the manuscript. We hope that the revised manuscript is now suitable for publication in Cancers.

Reviewer 3 Report

This article shows the potential correlation between changes in serum FGF21 levels and lenvatinib induced appetite loss in patients with unresectable hepatocellular carcinoma. It is interesting for me. However, there are several problems.

Major points:

1.     Serum FGF21 levels are affected by several factors including lifestyle behaviors (smoking and alcohol consumption, etc), age, and liver function. Was the FGF21 level at baseline high in patients with alcohol-related HCC? Is there any impact because of the rate of FGF21 change? The authors should describe this issue.

2.     Is it possible to calculate a cut-off value for Grade ≥ 2 lenvatinib-induced appetite loss at an increase in serum FGF21 level from base line to 4 weeks after treatment? The authors should describe this issue in detail.

3.     The authors should perform multivariate analysis for the factors, including FGF21 changes, associated with Grade ≥ 2 lenvatinib-induced appetite loss.

Minor points:

1.     The authors should describe the assessment criteria for adverse events (AEs).

2.     Best overall response was included in the baseline patient characteristics (Table 1). The authors should separate it. This part is a result.

3.     Figure 2 (A, B) was wrong.

4.     The authors should show each mark of FGF21 change from baseline in Figure 4.

Author Response

June 11, 2023

Prof. Dr. Samuel C. Mok

Editor-in-Chief

Cancers

Dear Editor:

We are grateful for your consideration of our manuscript titled “Potential correlation between changes in serum FGF21 levels and lenvatinib-induced appetite loss in patients with unre-sectable hepatocellular carcinoma.

We greatly appreciate the comments of the editor and reviewers and addressing them has significantly improved the quality of the manuscript. We hope that the revised manuscript is now suitable for publication in Cancers.

Our point-by-point responses to the editor and reviewers’ comments are attached below. Changes to the text are indicated in red for ease of review.

Thank you for your consideration. I look forward to hearing from you.

Sincerely,

Goki Suda, M.D., Ph.D

Department of Gastroenterology and Hepatology/Graduate School of Medicine

Hokkaido University, North 15, West 7, Kita-ku,

Sapporo, Hokkaido 060-8638, Japan

Phone number: +81 11-716-1161

Fax number: +81 11-706-7867

Email address: [email protected]

Responses to Reviewers

This article shows the potential correlation between changes in serum FGF21 levels and lenvatinib induced appetite loss in patients with unresectable hepatocellular carcinoma. It is interesting for me. However, there are several problems.

Response

We sincerely appreciate the reviewer’s helpful and constructive comments and suggestions. We have addressed each of these in detail below.

Major points:

  1. Serum FGF21 levels are affected by several factors including lifestyle behaviors (smoking and alcohol consumption, etc), age, and liver function. Was the FGF21 level at baseline high in patients with alcohol-related HCC? Is there any impact because of the rate of FGF21 change? The authors should describe this issue.

Response

We sincerely appreciate the reviewer’s helpful and constructive comments and suggestions. In this study, we could not collect some factors associated with changes in FGF21, including the data of alcohol intake during the lenvatinib treatment. Thus, we added this information to the limitations section of this study, in lines 337–339.

In addition, the FGF21 levels were similar between patients with or without alcohol-related HCC, as show below:

  1. Is it possible to calculate a cut-off value for Grade ≥ 2 lenvatinib-induced appetite loss at an increase in serum FGF21 level from base line to 4 weeks after treatment? The authors should describe this issue in detail.

Response

We sincerely appreciate the reviewer’s helpful and constructive comments and suggestions. We analyzed optimal cut-off values for the rate of change in FGF21 levels for predicting grade ≥ 2 appetite loss during lenvatinib treatment. As shown in Supplementary Figure S1, the cut-off value for the rate of change in FGF21 levels at 4 weeks after lenvatinib-treatment initiation for predicting grade ≥ 2 appetite loss was set at 0.174 (sensitivity 85.7, specificity 57.1; area under the ROC curve, 0.66 [95%CI 0.46–0.858]). We have added this description to lines 130–136, lines 248–252, and Supplementary Figure S1, .

  1. The authors should perform multivariate analysis for the factors, including FGF21 changes, associated with Grade ≥ 2 lenvatinib-induced appetite loss.

Response

We sincerely appreciate the reviewer’s helpful and constructive comments and suggestions. As pointed out by the reviewer, we conducted a multivariate logistic regression analysis with variables identified at P ≤ 0.15 in univariate analyses of the baseline factors associated with occurrence of grade ≥ 2 appetite loss during lenvatinib treatment, and the rate of change in FGF21 at 4 weeks after lenvatinib initiation. As shown in Supplementary Table S1, multivariate analysis revealed that the rate of change in FGF21 at 4 weeks after treatment with lenvatinib alone was significantly associated with occurrence of grade ≥ 2 appetite loss during lenvatinib treatment (odds ratio 1.010 [95%CI 1.001–1.020], p = 0.037) (Supplementary Table S1). We have added this description in lines 130–136, lines 253–257, and Supplementary Table S1.

Minor points:

  1. The authors should describe the assessment criteria for adverse events (AEs). Response

We sincerely appreciate the reviewer’s suggestions. We have added these points in lines 102–103.

  1. Best overall response was included in the baseline patient characteristics (Table 1). The authors should separate it. This part is a result.

Response

We sincerely appreciate the reviewer’s suggestions. As pointed out by the reviewer, we have separated these data and moved it to lines 151–153.

  1. Figure 2 (A, B) was wrong.

Response

We sincerely appreciate the reviewer pointing this out and have corrected it.

  1. The authors should show each mark of FGF21 change from baseline in Figure 4. 

Response

We sincerely appreciate the reviewer’s suggestions. We have revised Figure 4B.

We greatly appreciate the comments of the reviewers and addressing them has significantly improved the quality of the manuscript. We hope that the revised manuscript is now suitable for publication in Cancers.

References             

  1. Finn, R.S.; Kudo, M.; Cheng, A.L.; Wyrwicz, L.; Ngan, R.K.C.; Blanc, J.F.; Baron, A.D.; Vogel, A.; Ikeda, M.; Piscaglia, F., et al. Pharmacodynamic Biomarkers Predictive of Survival Benefit with Lenvatinib in Unresectable Hepatocellular Carcinoma: From the Phase III REFLECT Study. Clin Cancer Res 2021, 27, 4848-4858, doi:10.1158/1078-0432.CCR-20-4219.
  2. Yang, Z.; Suda, G.; Maehara, O.; Ohara, M.; Yoshida, S.; Hosoda, S.; Kimura, M.; Kubo, A.; Tokuchi, Y.; Fu, Q., et al. Changes in Serum Growth Factors during Lenvatinib Predict the Post Progressive Survival in Patients with Unresectable Hepatocellular Carcinoma. Cancers (Basel) 2022, 14, doi:10.3390/cancers14010232.
  3. Tillman, E.J.; Rolph, T. FGF21: An Emerging Therapeutic Target for Non-Alcoholic Steatohepatitis and Related Metabolic Diseases. Front Endocrinol (Lausanne) 2020, 11, 601290, doi:10.3389/fendo.2020.601290.
  4. Hughes, S.E. Differential expression of the fibroblast growth factor receptor (FGFR) multigene family in normal human adult tissues. J Histochem Cytochem 1997, 45, 1005-1019, doi:10.1177/002215549704500710.
  5. Lin, X.; Li, G.; He, X.; Ma, X.; Zhang, K.; Zhang, H.; Zeng, G.; Wang, Z. FGF21 inhibits apolipoprotein(a) expression in HepG2 cells via the FGFR1-ERK1/2-Elk-1 pathway. Molecular and Cellular Biochemistry 2014, 393, 33-42, doi:10.1007/s11010-014-2044-0.
  6. Zhang, Y.; Lei, T.; Huang, J.F.; Wang, S.B.; Zhou, L.L.; Yang, Z.Q.; Chen, X.D. The link between fibroblast growth factor 21 and sterol regulatory element binding protein 1c during lipogenesis in hepatocytes. Molecular and Cellular Endocrinology 2011, 342, 41-47, doi:https://doi.org/10.1016/j.mce.2011.05.003.

Round 2

Reviewer 3 Report

The revised article was well written according to reviewers’ comments. However, there are some problems in this article.

1.     Supplementary data showed the ROC curve analysis and multivariate analysis of factors associated with grade 2 or more appetite loss during lenvatinib-treatment for unresectable HCC. The AUC was 0.66 with a sensitivity of 85.7% and a specificity of 57.1%. Multivaiate analysis showed that the rate of changes in FGF21 at 4 weeks after lenvatinib were associated with occurrence of grate 2 or more appetite loss during lenvatinib (odds ratio, 1.010). The AUC and odds ratio are relatively low levels. The authors should describe the comments regarding this issue in the discussion section.

2.     The authors should plot each spot of baseline FGF21 in Fig. 4A and that of FGF change from baseline in Fig 4 C, D, E. Similarly, you should plot each spot of change in serum albumin levels from baseline in Fig.5.

Author Response

June 13, 2023

Prof. Dr. Samuel C. Mok

Editor-in-Chief

Cancers

Dear Editor:

We are grateful for your consideration of our manuscript titled “Potential correlation between changes in serum FGF21 levels and lenvatinib-induced appetite loss in patients with unre-sectable hepatocellular carcinoma.

We greatly appreciate the comments of the editor and reviewers and addressing them has significantly improved the quality of the manuscript. We hope that the revised manuscript is now suitable for publication in Cancers.

Our point-by-point responses to the editor and reviewers’ comments are attached below. Changes to the text are indicated in red for ease of review.

Thank you for your consideration. I look forward to hearing from you.

Sincerely,

Goki Suda, M.D., Ph.D

Department of Gastroenterology and Hepatology/Graduate School of Medicine

Hokkaido University, North 15, West 7, Kita-ku,

Sapporo, Hokkaido 060-8638, Japan

Phone number: +81 11-716-1161

Fax number: +81 11-706-7867

Email address: [email protected]

Responses to Reviewers

Reviewer 3

Comments and Suggestions for Authors

The revised article was well written according to reviewers’ comments. However, there are some problems in this article.

Response

We sincerely appreciate the reviewer’s helpful and constructive comments and suggestions. We have addressed each of these in detail below.

  1. Supplementary data showed the ROC curve analysis and multivariate analysis of factors associated with grade 2 or more appetite loss during lenvatinib-treatment for unresectable HCC. The AUC was 0.66 with a sensitivity of 85.7% and a specificity of 57.1%. Multivaiate analysis showed that the rate of changes in FGF21 at 4 weeks after lenvatinib were associated with occurrence of grate 2 or more appetite loss during lenvatinib (odds ratio, 1.010). The AUC and odds ratio are relatively low levels. The authors should describe the comments regarding this issue in the discussion section.

Response

We sincerely appreciate the reviewer for their insightful and constructive comments and suggestions. We are in complete agreement that the AUC regarding the rate of changes in FGF21 as a predictive factor, as well as the odds ratio in the multivariate analysis, are relatively low. Therefore, the predictive potential of the rate of changes in FGF21 is not considerably high. Considering the relatively low levels of AUC and odds ratio, it would be necessary to include additional predictive factors for appetite loss during lenvatinib treatment to improve the accuracy of the prediction model. Consequently, further research and analysis are indeed warranted. We have incorporated these comments into the revised manuscript in line 323 to 331

  1. The authors should plot each spot of baseline FGF21 in Fig. 4A and that of FGF change from baseline in Fig 4 C, D, E. Similarly, you should plot each spot of change in serum albumin levels from baseline in Fig.5.

Response

We sincerely appreciate the reviewer’s helpful and constructive suggestions. As pointed out by the reviewer, we revised Fig4A, C, D, E and Fig 5.

We greatly appreciate the comments of the reviewers and addressing them has significantly improved the quality of the manuscript. We hope that the revised manuscript is now suitable for publication in Cancers.

Round 3

Reviewer 3 Report

The revised manuscript is well responded according the reviewer’s comments. It’s ok in the revised manuscript.

Author Response

June 16, 2023

Prof. Dr. Samuel C. Mok

Editor-in-Chief

Cancers

Dear Editor:

We are grateful for your consideration of our manuscript titled “Potential correlation between changes in serum FGF21 levels and lenvatinib-induced appetite loss in patients with unre-sectable hepatocellular carcinoma.

We greatly appreciate the comments of the editor and reviewers and addressing them has significantly improved the quality of the manuscript. We hope that the revised manuscript is now suitable for publication in Cancers.

Our point-by-point responses to the editor and reviewers’ comments are attached below. Changes to the text are indicated in red for ease of review.

Thank you for your consideration. I look forward to hearing from you.

Sincerely,

Goki Suda, M.D., Ph.D

Department of Gastroenterology and Hepatology/Graduate School of Medicine

Hokkaido University, North 15, West 7, Kita-ku,

Sapporo, Hokkaido 060-8638, Japan

Phone number: +81 11-716-1161

Fax number: +81 11-706-7867

Email address: [email protected]

Responses to Reviewers

Reviewer 3

Comments and Suggestions for Authors

The revised manuscript is well responded according the reviewer’s comments. It’s ok in the revised manuscript.

Response

We greatly appreciate the comments of the reviewers and addressing them has significantly improved the quality of the manuscript. We hope that the revised manuscript is now suitable for publication in Cancers.
